# Fantastic Beasts and Why It Is Necessary to Understand Our Relationship—Animal Companionship under Challenging Circumstances Using the Example of Long-Covid

**DOI:** 10.3390/ani12151892

**Published:** 2022-07-25

**Authors:** Christine Krouzecky, Jan Aden, Katharina Hametner, Armin Klaps, Zuzana Kovacovsky, Birgit U. Stetina

**Affiliations:** 1Psychological Outpatient Clinic, Sigmund Freud University Vienna, 1020 Vienna, Austria; armin.klaps@sfu.ac.at (A.K.); zuzana.kovacovsky@sfu.ac.at (Z.K.); birgit.u.stetina@sfu.ac.at (B.U.S.); 2Faculty of Psychology, Sigmund Freud University Vienna, 1020 Vienna, Austria; jan.aden@sfu.ac.at (J.A.); katharina.hametner@sfu.ac.at (K.H.)

**Keywords:** “pet-effect paradox”, Long-Covid affliction, animal companionship, human–animal bond, COVID-19 pandemic, crisis response, stress

## Abstract

**Simple Summary:**

Our society seems to generate the general belief that caring for animals has a positive effect on the well-being of humans, which often creates unrealistic expectations in pet owners. Since there are contradictory results that underline the stressful features of the human–animal relationship, it seems to be highly relevant to investigate this phenomenon in more detail. The present study tries to examine how external stressors such as the COVID-19 pandemic as well as Long-Covid influence the biopsychosocial wellbeing of animal caregivers. The results demonstrate a gap between the subjectively experienced meaning of pets in the context of the pandemic, which is positive throughout, and the calculated findings from several psychometric instruments indicating that animal companionship might be seen as an additional burden in times of crisis. These results underline the existence of the so called “pet-effect paradox” which emphasizes the mismatch between pet owners’ individual perception regarding the importance of an animal and the measurable findings in research. Further research is needed to understand the underlying dynamics in more detail and to prevent false expectations in connection with the human–animal relationship.

**Abstract:**

Studies in the field of human–animal interaction tend to highlight the positive results of the influence of animals on humans, which supports the popular belief that the human–animal bond positively affects humans’ well-being (“pet-effect”). Nevertheless, contradictory results exist that seem especially visible since the COVID-19 pandemic, a prominent external stressor. Despite critical findings, individuals seem to want to believe in the beneficial effects of the human–animal relationship (“pet-effect paradox”). Based on this background, the present study aims to investigate this phenomenon using a mixed-method design. Therefore, animal caregivers were surveyed online and compared using psychometric measurements and open-ended questions. In this context, a special focus was placed on the additional stressor of Long-Covid and related concerns. The results demonstrate once more the existence of the “pet-effect paradox” due to a contradiction in the quantitative and qualitative results. At a quantitative level, the findings show additional burdens on animal caregivers who are confronted with multiple loads. However, the qualitative results indicate a belief in the beneficial effects of pets at the biopsychosocial level. Additionally, the data demonstrate a shift in focus away from the environment to oneself when affected by Long-Covid, which might affect the ability to care for an animal.

## 1. Introduction

Since the 1980s, studies in the field of human–animal interactions have suggested that relationships with companion animals are beneficial to humans on a bio-psycho-social level [1,2,3,4]. This assumption can be derived from various correlations between pet ownership and improved physical and psychological health as well as increased social functioning [5,6]. The resulting idea that owning a companion animal positively effects the well-being of humans has been named the “pet-effect”—a term that is still often used in anthrozoological research to describe the (alleged) beneficial relationship between individuals and animals [7]. One specific health promoting aspect for humans that has been considered in this context—especially by the media, but also in a variety of studies—is the human–animal bond as a potential source for a human’s wellbeing during stressful times [8]. In that regard, anthrozoological studies report that animal companionship can be seen as a moderator of stressful life events and that the relationship with pets has a general stress-reducing effect on humans, especially when coping with challenging situations [9]. Accordingly, findings indicate a connection between animal companionship and a lower probability of suffering from cardiovascular disease as well as enhanced oxytocin levels that positively influence humans stress reactivity [10,11]. Moreover, studies report that living with companion animals may benefit owners in terms of reduced feelings of loneliness through access to a perceived source of unconditional love, support, and stability [3,12]. These and many other relevant findings, which can be found in a large variety of publications, lead to the popular belief that living with companion animals positively influences a humans’ general well-being (i.e., the pet-effect).

Interestingly, in addition to this general conviction, contradictory results exist regarding the animal’s role in the context of stress management, but they receive little attention in public media as well as research [13,14]. In this context, studies suggest that animal companionship might be associated with increased symptoms of depression, higher levels of emotional distress, and a lower quality of life, especially during crises [15,16,17]. Moreover, the results indicate that, for example, during specific stressful situations such as the COVID-19 pandemic, pet owners report lower vitality, lower life satisfaction, and a lower life meaning, which might lead to the assumption that pets may contribute to an increased burden among owners [15,18].

Given these results, a question arises regarding how such contradictory findings are possible. The psychologist Hal Herzog emphasizes the so called “pet-effect paradox”, which explains a mismatch between pet owners’ individual perception regarding the importance of an animal—which seems to be predominantly positive—and the actual findings in research [19,20,21]. This includes the phenomenon that despite contradictory findings, individuals seem to *want* to believe in the beneficial effects of the human–animal relationship. The latest research results indicate divergencies between the opinions of pet owners regarding the pet’s protective role during the COVID-19 pandemic, focusing on their perception, and the results of standardized measurements [17]. At a quantitative level, the mentioned findings demonstrate that animal caregivers (compared to non-animal caregivers) report a significantly stronger expression of depressive symptoms as well as a significantly lower experienced quality of life using psychometric clinical instruments. Additionally, the results show that pet-related concerns and the strength of the measured human–animal bond seem to negatively influence the perception of one’s own wellbeing [17]. Yet, findings at a qualitative level indicate that animal caregivers reflexively perceive their animals as a positive influence on their biopsychosocial level during the pandemic [17].

From these results, it appears that the “paradox of the pet-effect” is particularly evident when individuals are confronted with different stressors. Therefore, the COVID-19 pandemic provided an opportunity for researchers in this field to examine this phenomenon with regard to a global stressor, affecting individuals worldwide, in more detail. In this context, most of the studies indicate a significant and complex influence of animal companionship and related responsibilities in the context of stress experience that seems to be strongly determined by external stressors [15,17,18]. Therefore, the present study aimed to investigate a wide range of external stressors that animal caregivers were exposed to during the last two years of the pandemic, including the long-term effects of a COVID-19 infection (“Long-Covid” or “Post COVID-19 condition”) [22] and concerns related to caring for an animal as well as the pandemic itself. Our research tries to examine the extent to which the bond and the responsibilities associated with owning an animal influence symptoms of depression and the quality of life and effects of social isolation (e.g., loneliness and social support) when confronted with above mentioned stressors. Using a mixed-methods design, pet owners were surveyed online and compared using the mentioned constructs.

## 2. Materials and Methods

### 2.1. Study Design

Using a cross-sectional mixed methods design, an online survey in the German language was conducted. The survey test battery included closed-ended and open-ended questions regarding demographic data and standardized questionnaires concerning the emotional attachment to the companion animal (Lexington Attachment to Pet Scale), symptoms of depression (Beck’s Depression Inventory), the quality of life (Quality of Life Questionnaire), loneliness (Loneliness Scale), and social support (ENRICHD Social Support Inventory). Pet owners were recruited via snowball sampling and surveyed online. A total of 242 data sets were fully completed (*n =* 242) and therefore included in statistical and qualitative analysis.

### 2.2. Instruments

The used test battery consisted of a series of standardized and open-ended questions using the mentioned instruments and strategies. In the following section, the psychometric instruments used are presented and open-ended questions are described.

#### 2.2.1. Demographic Data

Collected data included sex, age, highest educational attainment, job, current place of residence, nationality, marital status, living situation, and specifics about animal companionship such as species of the animal(s) or the duration of animal husbandry. Furthermore, questions regarding the current COVID-19 situation were asked including possible lockdowns, quarantines, and infections plus the long-term effects. With regard to a Long-Covid affliction, additional data were collected in relation to psychological and physical symptoms.

Overall, the sample consists of 83.8% women, 15.4% men, and 0.8% others. The average age of the sample was 36 years (M = 36.43; SD = 12.22). Regarding the marital status, the sample mainly consisted of those living in a relationship (33.8%), single (31.3%), or married (27.1%). Several participants stated that they were divorced (6.3%) or widowed (1.7%). About one third of the sample indicated having finished undergraduate or graduate school (bachelor’s or a master’s degree, 33.1%). A total of 17.6% stated having graduated from high school and 14.6% completed apprenticeship. To evaluate the current situation with regard to COVID-19, additional questions were asked about lockdown, quarantine, infection, and Long-Covid. In this context 32.7% of the sample stated having strict lockdown restrictions and 3.3% claimed to be in quarantine at the time of the survey. Regarding a COVID-19 infection, 56.5% stated that they were infected with COVID-19 at any time up to the survey and 50.3% indicated that they were suffering from Long-Covid at the time of the survey.

#### 2.2.2. Lexington Attachment to Pets Scale (LAPS)

The Lexington Attachment to Pets Scale [23] was used to measure the subjectively experienced strength of the human–animal bond. The questionnaire consists of 23 items that could be answered on a four-point Likert scale (from totally disagree to totally agree). The LAPS shows a good internal consistency (Cronbach alpha of α = 0.928). The evaluation was conducted by calculating the average value that represents the strength of emotional attachment to an animal subjectively experienced by its owner.

**Example statement:** *“I feel that my dog is part of my family”.*

#### 2.2.3. Beck Depression Inventory (BDI-II)

The BDI-II [24] was used to measure characteristic attitudes and symptoms of depression. It is a self-assessment inventory containing 21 items that can be answered on a four-point rating scale. Individuals are asked to rate the past seven days from the absence of a symptom to a strong manifestation. For evaluation, a sum score can be calculated, which represents the severity of depressive symptoms. The Beck Depression Inventory commands a good internal consistency (Cronbach alpha of α = 0.92).


**Example item “Sadness”:**
*“I do not feel sad” to “I am so sad and unhappy that I can’t stand it”.*


#### 2.2.4. Quality of Life Questionnaire (WHOQOL-BREF)

The WHOQOL-BREF [25] is a self-report rating inventory that was used for measuring the subjectively experienced quality of life. The questionnaire consists of 26 items that represent the five subscales/domains “Physical health”, Psychological health”, “Social relationships”, and “Environment”, which can be evaluated individually by mean values. Additionally, a total score can be calculated to evaluate the general experienced quality of life. Every item can be answered on a seven-point scale and scores are scaled in a positive direction (i.e., higher scores represent a higher quality of life). The internal consistency amounts to α = 0.88.


**Example question.**
*“How satisfied are you with your ability to perform your daily living activities?”*


#### 2.2.5. ENRICHD Social Support Inventory (ESSI)

The ESSI [26] was used to measure the subjectively experienced social support. The instrument contains six items that represent four defining attributes of social support: emotional, instrumental, informational, and appraisal, which are calculated by mean values. The response alternatives range between 1 (never) and 5 (always) and the internal consistency is α = 0.89.


**Example Statement.**
*“Is there someone available to you who you can count on to listen when you need to talk?”*


#### 2.2.6. Loneliness Scale

The Loneliness Scale [27] is a 20-item self-assessment inventory that was used to measure subjective feelings of loneliness as well as feelings of social isolation. Items can be answered on a four-point Likert scale from never (1) to often (4). For evaluation, a mean value can be calculated, whereby positively formulated items are recoded. Therefore, higher scores represent greater experienced feelings of loneliness. The internal consistency amounts to α = 0.94.


**Example Statement.**
*“I find myself waiting for people to call or write”.*


#### 2.2.7. Pet Related Stressors/Concerns

Specific pet related stressors/concerns were analyzed based on preliminary investigations [28]. In this context, a list of nine concerns that may affect caring for an animal during the pandemic has been provided, which can be rated on a five-point Likert scale (from totally disagree to totally agree).


**Example Statement.**
*“I’m worried that if I get sick, I won’t be able to take care of my pet.”*


#### 2.2.8. Pandemic-Related Stressors/Concerns

In addition to animal-related stressors, general stressors/concerns regarding the pandemic were surveyed based on previous studies [28]. In this context, a list of seven concerns that likely developed during the COVID-19 pandemic has been provided (e.g., concerns regarding social isolation or career future). Moreover, one additional item regarding concerns about travel restrictions has been added to the list but not included in statistical analysis of the present article due to its mismatch. All questioned concerns could be rated on a five-point rating scale (from totally disagree to totally agree).


**Example Statement.**
*“I worry about being socially isolated”.*


In addition to these closed-ended questions, an open-ended question (“What other concerns are on your mind regarding the corona virus (COVID-19)”) asked participants to raise other concerns that were not covered by the predefined items.

#### 2.2.9. Personal Opinion Regarding the Animals’ Role When Suffering from Long-Covid

Animal caregivers were asked about their personal opinion regarding the animals’ role during the pandemic in general and especially in connection with the management of Long-Covid in an open and closed question format. In this context, participants were asked about their agreement with seven pre-formulated statements using a four-point scale answer format from totally agree to totally disagree. Open questions were used to examine their individual opinions on the human–animal relationship during the pandemic and with regard to Long-Covid (e.g., “How does your pet impact the management of your Long-Covid condition”).

### 2.3. Statistical Analysis

Statistical analyses were computed with SPSS 27.0. Regarding pet-related and pandemic-related concerns, factor analyses were calculated based on the listed items. The aim was to examine latent dimensions and investigate underlying structures of individuals’ associations regarding evaluated concerns. Reliability was assessed with coefficient alpha.

Moreover, univariate procedures in the form of paired t-tests were chosen to gain insight into possible differences between animal-caregivers suffering from Long-Covid and animal-caregivers who have not been infected with the COVID-19 virus by the time of the survey with regard to biopsychosocial wellbeing (BDI-II, WHOQOL-BREF, ESSI, and Loneliness Scale), the human–animal bond (LAPS), and calculated factors of pet-related and pandemic-related concerns. Cohens d was calculated as an effect size measurement. Prior examination required conditions for t-tests and factor analyses were conducted.

Furthermore, correlations were calculated to measure mutual relations between pet-related and pandemic-related concerns, the strength of the human–animal bond (LAPS) and symptoms of depression (BDI-II), the subjectively experienced quality of life (WHOQOL-BREF), loneliness (Loneliness Scale), and social support (ESSI). For every analysis, the significance level was set at *p* ≤ 0.05.

### 2.4. Qualitative Analysis

In addition to statistical analyses, open-ended questions were examined using the qualitative content analysis according to Phillip A. E. Mayring [29]. Individual concerns related to the pandemic or to the COVID-19 virus as well as the personal opinion of pet owners regarding the animal’s role during the pandemic in general and especially with regard to the management of Long-Covid showed the potential for systematic classification. Therefore, individual concerns regarding the pandemic or the COVID-19 virus were categorized based on a previous study that examined three main categories and twelve subcategories of pandemic-related concerns with the help of a factor analysis [30] (=deductive categorization). In this context, answers were individually coded by breaking down the relevant text into short strings of words, capturing the meaning of participants’ expressions, and assigning answers/expressions to defined categories. Since the analysis of the data gave an indication that certain content needed a different categorization than what was available, three subcategories were added to the category system. These are described in more detail as part of the presentation of the results.

The deductive categorization was also chosen for the thematic analysis of personal opinions regarding the animal’s role during the pandemic in general and especially regarding the management of Long-Covid. Thus, deductive categories were developed based on the theoretical basis of the bio-psycho-social model. To assure reliability, the coding process was carried out by two experts independently and analyses were subjected to a reliability test (Cohens Kappa > 0.60).

## 3. Results

### 3.1. Statistical Analysis

#### 3.1.1. Pet-Related and Pandemic-Related Concerns

To investigate the underlying structures of the assessed pet-related and pandemic-related concerns, exploratory factor analyses were calculated. Table 1 represents the dimensional reduction of the pet-related concerns and Table 2 represents the dimensional reduction of the pandemic-related concerns.

The exploratory factor analysis regarding pet-related concerns resulted in a two-factor solution: responsibility-related concerns and animal-related concerns. All of the initial nine items were retained. The first factor “responsibility-related concerns” describes concerns associated with no longer being able to adequately care for the pet for various reasons. This factor consists of five items (No. 2, 3, 5, 6, and 7) and has a very good reliability (α = 0.855). The second factor “pet-related concerns” describes concerns that explicitly refer to the well-being of the animal. It consists of four items (No. 1, 4, 8, and 9) and has a good reliability (α = 0.805).

The exploratory factor analysis regarding pandemic-related concerns also resulted in a two-factor solution: general pandemic-related concerns and infection-related concerns. As previously mentioned, only seven of the eight items have been included in the calculations, as one independent item has been added to the existing list to gain more information. Nevertheless, this item was excluded in the present analysis to avoid distorting the scale calculations. The first factor, “general pandemic-related concerns”, describes concerns associated with daily worries related to the overall situation of the pandemic. This factor consists of five Items (No. 3, 4, 5, 6, and 7) and has a moderate reliability (α = 0.680). The second factor, “infection-related concerns”, describes concerns that explicitly refer to one’s own infection with COVID-19 or the infection of a close person. It consists of two items (No. 1 and 2) and has a very good reliability (α = 0.878).

For a better understanding, Figure 1 and Figure 2 show the average expression of the evaluated concerns of animal caregivers during the pandemic (pet-related and pandemic-related).

To determine relationships between pet-related and pandemic-related concerns, Pearson correlations were conducted between the pre-calculated factors of pet-related concerns (animal-related concerns and responsibility-related concerns) and pandemic-related concerns (infection-related concerns and general pandemic-related concerns). Table 3 represents the calculations.

The results show significant positive correlations between animal-related concerns and responsibility-related concerns as well as between infection-related concerns and general pandemic-related concerns. Moreover, a significant negative correlation was found between responsibility-related concerns and infection-related concerns to the effect that the more infection-related concerns are pronounced the less responsibility-related concerns are perceived, or the other way around.

#### 3.1.2. Differences between the Groups “Long-Covid” and “No Infection” in Regard to the Biopsychosocial Wellbeing, the Human–Animal Relationships, and Concerns (Pet-Related as Well as Pandemic-Related)

In order to test whether the additional stressor of a chronic disease such as a Long-Covid affliction influences the biopsychosocial wellbeing of animal caregivers during the pandemic, participants suffering from Long-Covid and participants who had not been infected with the COVID-19 virus by the time of the survey were compared via t-test analysis. Additionally, the possible differences were analysed regarding the strength of the human–animal bond and regarding pet-related and pandemic-related concerns. Table 4 represents the results of the calculations.

The data show that there are significant differences between animal caregivers suffering from Long-Covid and animal caregivers who have not been infected with the COVID-19 virus by the time of the survey regarding the biopsychosocial wellbeing (symptoms of depression, all areas of the quality of life, and the perception of social support). In this context, the data demonstrate that animal caregivers suffering from Long-Covid report more symptoms of depression or a stronger expression of these symptoms, a significantly lower experienced quality of life in all the evaluated areas, and a lower sense of social support.

Moreover, the data show significant differences between the two samples regarding the assessed strength of the human–animal relationship and pet-related as well as pandemic-related concerns. In this connection, animal caregivers suffering from Long-Covid address a significantly stronger experienced relationship to the animal and significant less distinctive concerns regarding the pet. Furthermore, they report significantly stronger experienced concerns concerning COVID-19 infections and the pandemic in general. No significant differences were analyzed with regard to the perceived feeling of loneliness.

In order to investigate the previously mentioned differences in more detail and to gain insight into concerns as possible influencing factors, Pearson correlations between the two factors of pet-related concerns (animal-related concerns and responsibility-related concerns) and the variables “strength of the human–animal relationship” (LAPS), “symptoms of depression” (BDI-II), “quality of life” (WHOQOL-BREF), “loneliness” (Loneliness Scale), and “social support” (ESSI) were calculated separately in the groups “Long-Covid” and “No Infection”. The same procedure has been conducted with respect to the two factors of pandemic-related concerns (infection-related concerns and general pandemic-related concerns).

#### 3.1.3. Influence of Pet-Related Concerns on the Biopsychosocial Wellbeing and the Human–Animal Relationship in the Groups “Long-Covid” and “No Infection”

The results within the group “Long-Covid” show significant negative correlations between the factor “animal-related concerns” and the overall experienced quality of life, as well as the subdomains “environment” and “social relationships”. Moreover, regarding the factor “responsibility-related concerns”, a significant negative correlation was found with the overall experienced quality of life. Table 5 represents these significant results according to the calculations.

No significant results were found between the factor “animal-related concerns” and the variables “psychological health” (*r*(73) = −0.173, *p* = 0.143), “physical health” (*r*(61) = −0.157, *p* = 0.178), “depressive symptoms” (*r*(55) = 0.252, *p* = 0.064), “loneliness” (*r*(79) =0.160, *p* = 0.160), “social support” (*r*(81) = −0.108, *p* = 0.338), and “human–animal relationship” (*r*(78) = 0.212, *p* = 0.063). Additionally, no significant results were found between the factor “responsibility-related concerns” and “psychological health” (*r*(73) = −0.184, *p* = 0.120), “environment” (*r*(73) = −0.208, *p* = 0.077), “social relationships” (*r*(75) = −0.156, *p* = 0.182), “physical health” (*r*(75) = −0.147, *p* = 0.209), “depressive symptoms” (*r*(55) = 0.134, *p* = 0.328), “loneliness” (*r*(79) = −0.025, *p* = 0.827), “social support” (*r*(81) = −0.114, *p* = 0.311), and “human–animal relationship” (*r*(77) = 0.207, *p* = 0.071).

The results within the group “No Infection” show a significant positive correlation between the factor “animal-related concerns” and the variable “human–animal relationship”. Regarding the factor “responsibility-related concerns”, the results also demonstrate a significant positive correlation with the variable “human–animal relationship”. Table 6 represents these significant results.

No significant results were found between the factor “animal-related concerns” and the variables “psychological health” (*r*(62) = −0.224, *p* = 0.080), “environment” (*r*(61) = −0.188, *p* = 0.146), “social relationships” (*r*(61) = 0.082, *p* = 0.530), “physical health” (*r*(61) = −0.133, *p* = 0.307), “overall quality of life” (*r*(60) = −0.115, *p* = 0.381), “depressive symptoms” (*r*(53) = 0.153, *p* = 0.274), “loneliness” (*r*(52) = 0.206, *p* = 0.143), and “social support” (*r*(54) = −0.142, *p* = 0.306). Moreover, no significant correlations were found between the factor “responsibility-related concerns” and the variables “psychological health” (*r*(64) = −0.158, *p* = 0.212), “environment” (*r*(63) = −0.123, *p* = 0.337), “social relationships” (*r*(63) = −0.103, *p* = 0.423), “physical health” (*r*(63) = −0.143, *p* = 0.262), “overall quality of life” (*r*(62) = -.0165, *p* = 0.200), “depressive symptoms” (*r*(55) = 0.040, *p* = 0.774), “loneliness” (*r*(54) = 0.131, *p* = 0.346), and “social support” (*r*(55) = −0.160, *p* = 0.243).

#### 3.1.4. Influence of Pandemic-Related Concerns on the Biopsychosocial Wellbeing and the Human–Animal Relationship in the Groups “Long-Covid” and “No Infection”

The results within the group “Long-Covid” demonstrate significant negative correlations between the factor “infection-related concerns” and the variables “psychological health”, “physical health”, and “overall quality of life”. Additionally, a positive correlation was found with the variable “symptoms of depression”. Moreover, the data show significant negative correlations between the factor “general pandemic-related concerns” and the variables “psychological health”, “environment”, “social relationships”, and “overall quality of life”. Positive correlations were found regarding the variables “symptoms of depression” and “loneliness”. Table 7 represents these significant correlations.

No significant correlations were found between the factor “infection-related concerns” and the variables “environment” (*r*(73) = −0.187, *p* = 0.113), “social relationships” (*r*(75) = −0.058, *p* = 0.619), “loneliness” (*r*(79) = 0.066, *p* = 0.561), “social support” (*r*(81) = −0.105, *p* = 0.345), and “human–animal relationship” (*r*(79) = 0.127, *p* = 0.263). Additionally, no significant correlations were found between the factor “general pandemic-related concerns” and “physical health” (*r*(69) = −0.183, *p* = 0.133), “social support” (*r*(75) = −0.129, *p* = 0.271), and “human–animal relationship” (*r*(73) = −0.019, *p* = 0.875).

The results within the group “No Infection” show a significant negative correlation between the factor “infection-related concerns” and the variable “physical health” and a significant positive correlation with the variable “loneliness”. Regarding the factor “general pandemic-related concerns”, the results demonstrate significant negative correlations with the variables “psychological health”, “environment”, “social relationships”, “physical health”, “overall quality of life”, and “social support”. Additionally, positive correlations were found with the variables “symptoms of depression” and “loneliness”. Table 8 represents these significant results according to the calculations.

No significant correlations were found between the factor “infection-related concerns” and the variables “psychological health” (*r*(66) = −0.1651, *p* = 0.196), “environment” (*r*(65) = −0.170, *p* = 0.177), “social relationships” (*r*(65) = −0.020, *p* = 0.875), “overall quality of life” (*r*(64) = −0.198, *p* = 0.117), “depression” (*r*(57) = −0.159, *p* = 0.237), “social support” (*r*(56) = 0.027, *p* = 0.844), and “human–animal relationship” (*r*(76) = 0.024, *p* = 0.836). Moreover, no significant correlations were found between the factor “general pandemic-related concerns” and the variable “human–animal relationship” (*r*(74) = −0.161, *p* = 0.169).

To investigate the possible influence of the human–animal relationship on biopsychosocial parameters, Pearson correlations were conducted between the strength of the human–animal relationship (LAPS) and “symptoms of depression” (BDI-II), “quality of life” (WHOQOL-BREF), “loneliness” (Loneliness Scale), and “social support” (ESSI).

#### 3.1.5. Influence of the Strength of the Human–Animal Relationship on the Biopsychosocial Wellbeing in the Groups “Long-Covid” and “No Infection”

The results within the group “Long-Covid” show significant negative correlations between the variables “human–animal relationship”, “psychological health”, and “overall quality of life” as well as a positive correlation with the variable “symptoms of depression”. Table 9 represents these significant correlations.

No significant correlations were found between the variables “human–animal relationship” and “environment” (*r*(65) = −0.167, *p* = 0.182), “social relationships” (*r*(68) = −0.177, *p* = 0.150), “physical health” (*r*(67) = −0.187, *p* = 0.130), “loneliness” (*r*(70) = −0.054, *p* = 0.660), and “social support” (*r*(72) = −0.174, *p* = 0.144).

The results within the group “No infection” overall show no significant correlations between the variables “human–animal relationship” and the calculated parameters, i.e., “psychological health” (*r*(64) = −0.065, *p* = 0.611), “environment” (*r*(64) = −0.085, *p* = 0.505), “social relationships” (*r*(64) = 0.169, *p* = 0.181), “physical health” (*r*(63) = −0.062, *p* = 0.628), “overall quality of life” (*r*(63) = 0.010, *p* = 0.937), “symptoms of depression” (*r*(55) = 0.078, *p* = 0.572), “loneliness” (*r*(55) = 0.023, *p* = 0.867), and “social support” (*r*(55) = −0.020, *p* = 0.887),.

### 3.2. Qualitative Analysis

As it was stated before, the individual perception of concerns and the individual meaning of one’s pet during the pandemic was explored using open question format in addition to the rated items. The results of the quantitative content analysis according to Mayring (2015) are presented as follows.

#### 3.2.1. Reflexive Self-Perception of Concerns during the Pandemic

Regarding the reflexive self -perception of concerns during the COVID-19 pandemic, the open question “What other concerns are on your mind regarding the Corona virus (COVID-19)” was asked. The method of deductive categorization was used to structure the data. In this context, the participants’ answers were assigned to three main categories and twelve subcategories of an already existing concern scheme according to previous research [23]. Since the analysis provided an indication that additional categories were needed to include all the important content in the evaluation, the subcategory “other concerns related to social relations” was added to the existing concern scheme. Moreover, the subcategory “own health” has been differentiated into “own health—physiological” and “own health—psychological”. Table 10 represents the developed categories.

To find out if and how the subjective concerns of the participants differed depending on whether or not they were suffering from a Long-Covid affliction, the responses were analyzed separately within the groups “Long-Covid” and “No infection”. Figure 3 and Figure 4 represent the frequency analysis in percentages.

The results of the frequency analysis show that animal caregivers who were suffering from a Long-Covid affliction by the time of the survey mostly stated concerns that could be assigned to the category “own health—physical” (44.87%). Regarding animal caregivers without an infection by the time of the survey, the results of the frequency analysis demonstrate that concerns are most frequently assigned to the subcategories “social division” (23.91%), “restrictions imposed by fundamental rights and freedoms” (19.57%), and “health of related persons” (17.93%).

#### 3.2.2. Personal Opinion of the Animal’s Role during the Pandemic and Regarding Coping with Long-Covid

To investigate the subjective opinion regarding the animal’s role during the pandemic in general as well as in connection with coping with Long-Covid, the following open questions were asked: “What does/do your pet(s) mean to you during the COVID-19 pandemic” and “How does/do your pet(s) impact your Long-Covid condition?”. The deductive development of the categories resulted in a three-part categorization including the categories “Biological impact”, “Psychological impact”, and “Social impact”, which were proved to allow for the most appropriate categories to encompass the content of open answers. Table 11 represents the developed categories.

Again, the content of the answers was analyzed separately within the groups “Long-Covid” and “No infection” to investigate the possible differences. Figure 5 and Figure 6 represent the frequency analysis in percentages.

The results of the frequency analysis show that the subjective statements of animal caregivers who were suffering from a Long-Covid affliction by the time of the survey, in relation to their pets’ role during the COVID-19 pandemic, are most frequently assigned to the category “Psychological impact” (60.0%). In this context, the data show the highest frequency regarding the category with positive characteristics (42.5%). A total of 10% percent of the analyzed data included neutral characters and 7.5% included negative characters within the statements. These results are followed by the category “Biological impact”, where the 27.5% positively characterized, 17.5% neutrally characterized, and 7.5% negatively characterized statements were analyzed. Moreover, the category “Social impact” shows the lowest least content allocation and demonstrates 20% positively characterized and 7.5% neutrally characterized statements.

Within the group “No infection”, the frequency analysis demonstrates that the stated role of pet(s) during the pandemic could mostly be assigned to the category “Psychological impact” (39.6%). The data show the highest frequency concerning the category with positive characteristics (32.1%) and 7.5% of statements were categorized as neutral statements. These results are followed by the category “Social impact” where the 22.6% positively characterized and 9.3% neutrally characterized statements were analyzed. The category “Biological impact” included 9.4% positively characterized and 3.7% neutrally characterized statements.

## 4. Discussion

The present study aims to investigate the influence of animal companionship on different biopsychosocial parameters when one is confronted with multiple external loads such as the COVID-19 pandemic and the additional stressor of a Long-Covid affliction. A special focus was placed on concerns, which were evaluated both as being pet-related and pandemic-related. In this context, the data overall show that there is no evidence of the often-reported animals’ protective role regarding the biopsychosocial wellbeing at a quantitative level. On the contrary, the data might indicate an additional burden caused by the human–animal relationship during the pandemic and especially when one is confronted with multiple stressors. Pandemic-related concerns seem to have a particularly important influence on the biopsychosocial wellbeing of animal caregivers suffering from Long-Covid. Nevertheless, at a qualitative level, the data demonstrate that animal caregivers in both groups (“Long-Covid” and “No infection”) mostly estimate their animals as a biopsychosocial source during the pandemic. For a better understanding, the detailed results are discussed after being separated into two groups.

### 4.1. The Meaning of Concerns in Connection with Animal Companionship of Individuals Suffering from Long-Covid Condition

Within the group of animal caregivers suffering from a Long-Covid affliction, the data overall indicate a vulnerability regarding biopsychosocial health. The results demonstrate that affected participants reported significantly more or stronger experienced symptoms of depression as well as a significantly lower quality of life than animal caregivers without an infection by the time of the survey. Moreover, animal caregivers suffering from Long-Covid stated a significantly lower experienced feeling of being socially supported. These results are consistent with previous findings indicating the impaired physical and mental health of individuals with Long-Covid [31,32]. In addition to the obvious explanations for the decreased health status of Long-Covid sufferers, the present results provide evidence that other possible influencing factors are health-related concerns that might have a substantial impact on the well-being of affected animal caregivers. In this context, the findings at a quantitative level demonstrate that animal caregivers suffering from Long-Covid stated significantly stronger pronounced pandemic-related concerns than animal caregivers without an infection. This finding leads to the assumption that affected animal caregivers are more concerned about their own health along with the health of related persons. They also experience more daily worries related to the COVID-19 pandemic, which is supported at a qualitative level. Individuals affected by Long-Covid stated much higher pronounced concerns overall but especially concerning their own health at a physical as well as on a psychological level than animal caregivers without infection. Having a closer look into answers regarding health-related concerns, statements such as “I am afraid to be dependent on professional support in the near future because of long-term consequences” and “I am worried that my Post-Covid complaints will never go away“ suggest that individuals suffering from Long-Covid experience limited beliefs in overcoming the disease. Therefore, it might be assumed that the confrontation with a stressor such as a little researched chronical disease like Long-Covid would make a person less likely to believe in performing behaviors necessary to cope with the disease. This is also supported by the latest studies that have determined significant correlations between self-efficacy and the perception of disruption in the context of a COVID-19 infection [33]. Furthermore, the findings indicate that the confrontation with one’s own vulnerability leads to a higher perception of stress influenced by a decrease in self-efficacy [33]. In terms of social psychology, these results provide evidence that an infection with COVID-19 and its associated long-term effects can be seen as existential stressors or critical life events that might lead to a shock in the basic assumption that the world is benevolent, meaningful, and that the self is worthy (= shattered assumptions theory) [34].

With regard to the animal’s role during the pandemic and especially when one is confronted with an additional stressor such as a Long-Covid, the findings of the present study once again indicate that there is no sufficient evidence for the often-reported supportive influence of animal companionship. The results at a quantitative level show that animal caregivers suffering from Long-Covid experience a significantly stronger relationship with their animals, but this stronger experienced bond does not seem to have a positive influence on the wellbeing of their owners. On the contrary, the data demonstrate that within the sample of animal caregivers suffering from Long-Covid, a stronger evaluated relationship with the animal is associated with significantly lower levels of the quality of life, psychological health and with higher levels of depressive symptoms. Additionally, no indication could be found that a stronger perceived human–animal relationship influences concerns (positively or negatively) regarding the pandemic. These results suggest that the often-reported positive effects of animals on humans’ wellbeing are not visible on a measurable level within the present study. Nevertheless, on a reflexively perceived qualitative level, companion animals seem to be an important source with respect to the biopsychosocial wellbeing, especially when one is confronted with external stressors. Most animal caregivers suffering from Long-Covid stated a positive influence of the animal especially on a psychological level but also on a biological and social level. Even though some negative statements including negative characteristics such as “It is exhausting to go for a walk with the two twice a day even if you have no power”, the overall reflexive perception of the animal’s influence turns out positive.

Since the measurable findings suggest that the participants affected by Long-Covid are suffering from a significantly lower biopsychosocial wellbeing, which can even be linked to the relationship with an animal, it can be assumed that this gap between the qualitative and quantitative findings is a result of the previously described “pet-effect paradox” [19,20,21]. According to some critical statements of animal caregivers suffering from Long-Covid regarding their pet’s impact on the management of Long-Covid, it nevertheless seems that the “pet-effect paradox” is at least somewhat put into perspective when people are confronted with their own vulnerability. One potential explanation for this finding is the previously mentioned focus of affected individuals on their own physical and psychological health, which might cause a shift in their priorities and lead to changes regarding the possibility of caring for others. This assumption is also supported by the measurable findings underlining that animal caregivers suffering from Long-Covid stated significantly lower concerns regarding their pets and significantly higher concerns regarding COVID-19 infection and the pandemic in general. Moreover, a negative correlation between responsibility-related concerns and infection-related concerns was found, which might indicate that the stronger pronounced worries about oneself or close persons are, the less focus there is on the responsibility towards the animal. In this context, the data additionally indicate that pet-related concerns are not influenced by the strength of the human–animal bond but by parameters of the own wellbeing, which is shown by the negative correlations between the quality of life and animal-related as well as responsibility-related concerns. Therefore, it might be assumed that the confrontation with an existential stressor leads to an inward focus and to a reduction in responsibility-related aspects such as concerns around the animal.

### 4.2. The Meaning of Concerns in Connection with the Human–Animal Relationship within the Group “No Infection”

Within the group of animal caregivers without a COVID-19 infection, the data overall indicate that individuals seem to experience a better biopsychosocial wellbeing than animal caregivers suffering from Long-Covid. In this context, as previously stated, the results at a quantitative level demonstrate significant differences between the groups “Long-Covid” and “No Infection” regarding the biopsychosocial health. The only exception in this context is the feeling of loneliness, which does not differ between the two groups. This result might underline the possibility that individuals living with a companion animal experience them as a social source that is not influenced by an impairment of the caregiver’s health.

Furthermore, the data within this group overall demonstrate that animal caregivers without an infection seem to be more concerned about others (including the animal) than themselves. In this context, the results show significantly higher experienced worries regarding the pet and significantly lower pronounced concerns regarding the pandemic when compared with animal caregivers suffering from Long-Covid. Moreover, the data at a qualitative level indicate that the most pronounced worries during the pandemic are concerns regarding social division followed by concerns on the health of related persons, restrictions imposed on fundamental rights and freedoms, and other everyday problems in the context of the pandemic. Looking into the comparison with animal caregivers suffering from Long-Covid, these results might indicate that not being confronted with multiple stressors or acute stressors such as the Long-Covid and related health issues leads to an expansion of concerns away from the individual to the social environment.

Regarding the human–animal relationship, no evidence was found that the bond with an animal influences animal caregivers’ wellbeing positively. No significant correlation was found between the strength of the human–animal relationship and any of the biopsychosocial parameters. Therefore, it might be assumed that the aspect of animal companionship alone is not linked to wellbeing during the pandemic when not confronted with an additional stressor. These results are consistent with previous findings that suggest there is no significant influence of pet ownership on the mental or physical health during the COVID-19 pandemic [13]. In contrast, the results within the group of animal caregivers without an infection tend to give another indication that the relationship with an animal is more likely to create stress. The findings show that the stronger the relationship with an animal is perceived, the more the pet-related concerns are pronounced. Nevertheless, at a qualitative level, animal caregivers within this group reflexively perceive their pets as an important support on a biopsychosocial level when they are asked about the subjective meaning of the animal during the pandemic. The results show even more positive expressions regarding the subjectively experienced supportive role of animals, compared to animal caregivers suffering from Long-Covid. Therefore, we propose that according to the pet-effect paradox [19,20,21], individuals within this group want to believe in the positive support of their animals.

## 5. Limitations

One limiting factor of the present study is that the sample was biased towards highly educated women as a result of survey recruitment strategies. A more diverse sample might generate results that were not apparent with this sample; however, we chose the random sample to prioritize the speed of response and to gather new information as the COVID-19 pandemic emerged and spread across Austria and Germany. Another critical aspect is that psychometric instruments such as the ones used in the present study might reach their limits due to the extraordinary situation, i.e., a global crisis. Therefore, mixed-methods approaches especially facilitate the generation of hypotheses. In connection with the used qualitative methodological approach, it must be stated that open questions could be widened as the question about “other” concerns might have limited individual answers. Additional interviews examining the reflexive self-perception in relation to the influence of animals during crises could provide more practical insights. This aspect has been considered for our future research. Besides these limitations, our present findings underline the need for further studies that examine the aspect of the contradiction between the inducted ideas regarding the benefits of animals and the actual measurable findings.

## 6. Conclusions

Overall, the results of the present study indicate the relevant influence of stressors and related concerns on the biopsychosocial wellbeing of animal caregivers. The statistical results demonstrate that multiple external loads (e.g., COVID-19 and Long-Covid) lead to significantly lower levels of quality of life and higher levels of depression. Moreover, the findings suggest that animal companionship may cause an additional burden if someone is confronted with these stressors. Significant correlations were found between higher scores of the subjectively experienced strength of the human–animal relationship and lower levels of the biopsychosocial wellbeing as well as higher levels of psychological distress. Despite these data, the qualitative results, which include reflexive perceptions on the role of pets during the pandemic, paint a very different picture, with almost exclusively positive statements about the importance of the pet. Therefore, a discrepancy between the personal subjective view regarding the animal and the measured results has been identified once more, suggesting that the “pet-effect paradox” is particularly visible when animal caregivers are confronted with various external stressors.

Additionally, the results of the present study indicate that the confrontation with existential and acute stressors such as Long-Covid leads to an increased focus on one’s own person and a reduction of the focus on the animal, which might affect the ability to care for an animal in a similar fashion to before the infection. This aspect should be considered in future research and especially in public media since, despite such critical results, there are still recommendations for all people to adopt animals because of their health promoting effects [35]. The indications in this direction seem highly relevant to protect animal caregivers from false expectations when they enter into a human–animal relationship, which in turn can also protect animals, for example, from being abandoned.

Although some statistical results of the present study contradict the existing positive image of pet ownership, the subjective importance of animals for their caregivers cannot be ignored. We propose a closer analysis of the “pet-effect paradox” for a better understanding of how this phenomenon occurs.

## Figures and Tables

**Figure 1 animals-12-01892-f001:**
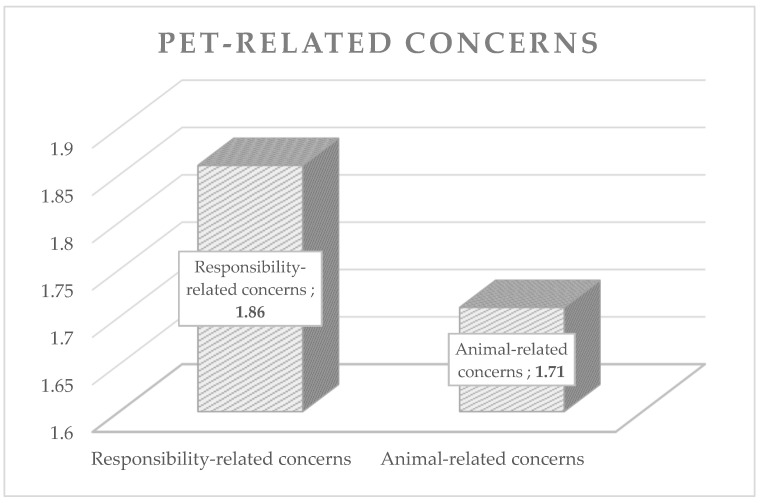
Mean values of pet-related concerns.

**Figure 2 animals-12-01892-f002:**
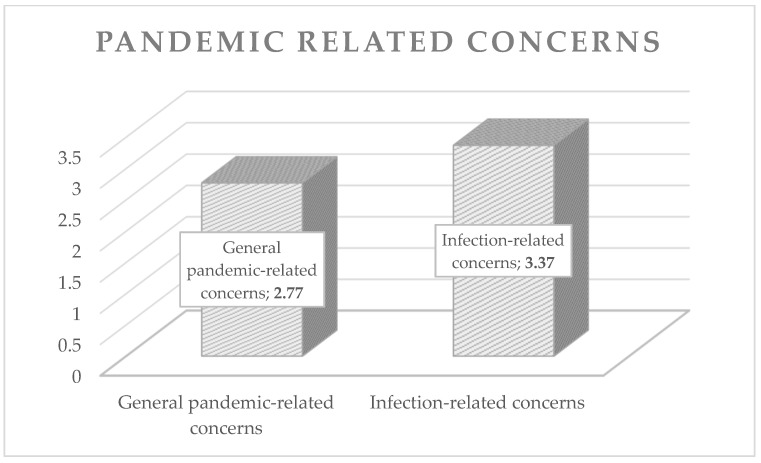
Mean values of pandemic-related concerns.

**Figure 3 animals-12-01892-f003:**
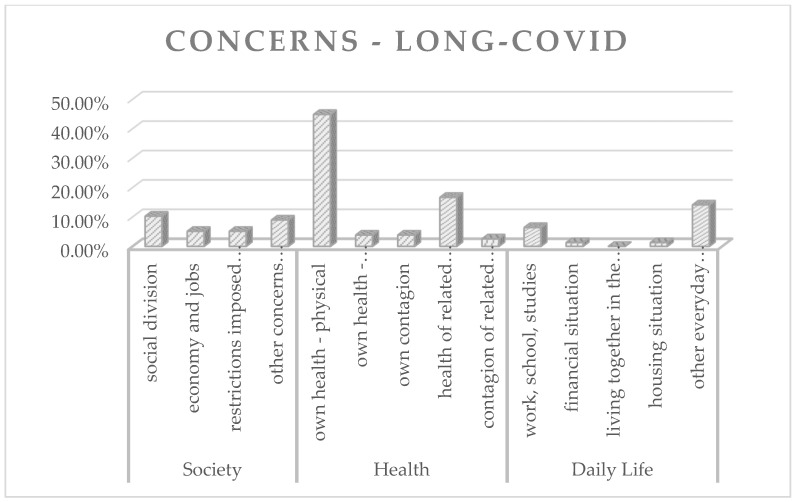
Percentages of pandemic-related concerns within the group “Long-Covid”.

**Figure 4 animals-12-01892-f004:**
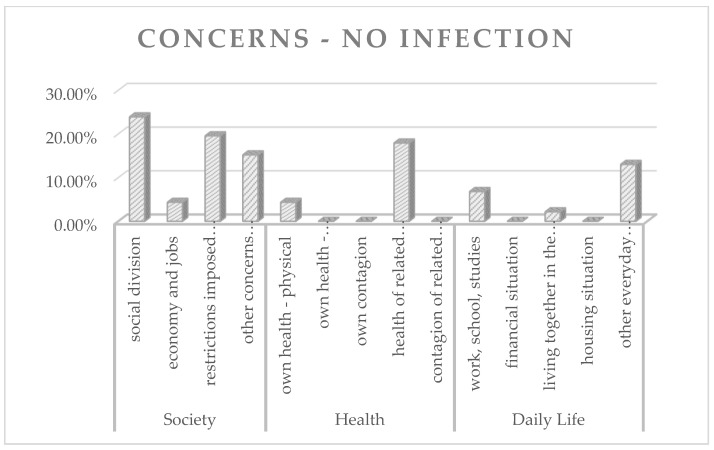
Percentages of pandemic-related concerns within the group “No infection”.

**Figure 5 animals-12-01892-f005:**
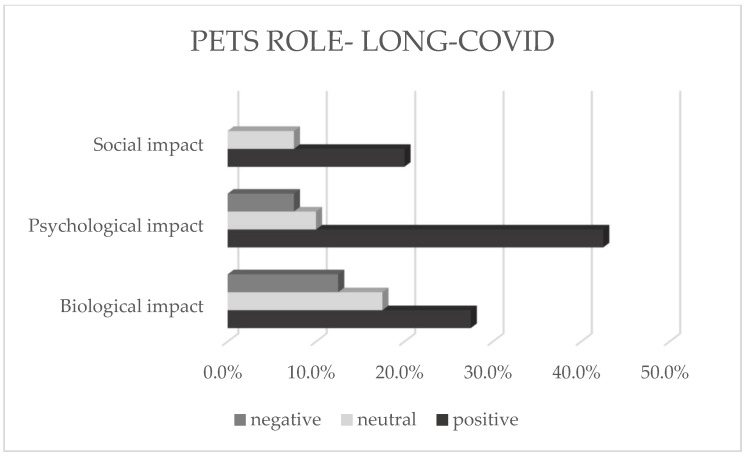
Percentages of the pet’s role within the group “Long-Covid”.

**Figure 6 animals-12-01892-f006:**
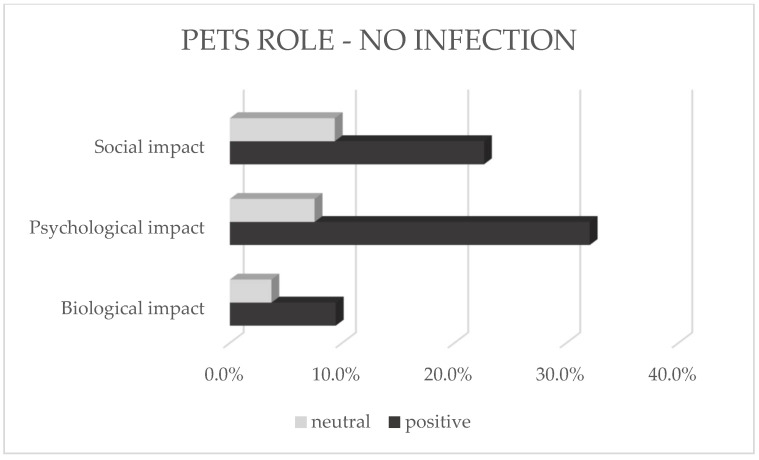
Percentages of the pet’s role within the group “No infection”.

**Table 1 animals-12-01892-t001:** Factor analysis of pet-related concerns.

	M	SD	Factor Loading	
**Item**			**Factor 1** (responsibility-related concerns)	**Factor 2** (animal-related concerns)
I am worried that i will not be able to take good care of my pet (buy food, take it for a walk, etc.) since I don’t leave the house anymore or rarely.	1.91	1.251	**0.790**	0.350
I’m worried that if I get sick, I won’t be able to take care of my pet.	2.33	1.419	**0.673**	0.439
I am worried that the daily routine between me and my pet will change negatively (not going out anymore etc.).	1.64	1.119	**0.740**	
I am worried that I will no longer have time for my pet due to career changes.	1.70	1.230	**0.772**	
I am worried that due to financial changes i will no longer be able to keep my pet.	1.57	1.093	**0.724**	0.325
I am worried that my pet will sense my fears and I will transfer them to my pet.	1.73	1.141	0.411	**0.708**
I am worried that the situation will have a negative impact on my pet’s health (vets are closed/going to the vet is not possible etc.).	1.80	1.264	0.418	**0.692**
I am worried that my pet will be unwell due to the isolation (no more contact with other animals and/or family members).	1.51	1.006	0.411	**0.600**
I am worried that my pet might get infected.	1.66	1.084		**0.832**

**Table 2 animals-12-01892-t002:** Factor analysis of pandemic related concerns.

	M	SD	Factor Loading	
**Item**			**Factor 1** (general pandemic-related concerns)	**Factor 2** (infection-related concerns)
I am worried about my professional future.	2.54	1.509	**0.510**	0.352
I am worried about the care of my children.	1.66	1.190	**0.497**	
I worry about being socially isolated.	2.62	1.364	**0.720**	
I am concerned about the economic situation.	3.30	1.296	**0.784**	
I am worried about social cohesion.	3.51	1.295	**0.685**	
I am worried about getting infected with the Corona virus.	2.99	1.430		**0.925**
I am worried that my family/friends will be infected with the Corona virus.	3.59	1.356		**0.918**

**Table 3 animals-12-01892-t003:** Correlations between pet-related and pandemic related concerns.

		Animal-Related Concerns	Responsibility-Related Concerns	Infection-Related Concerns	General Pandemic-Related Concerns
animal-related concerns	*r*	1	0.650	0.042	0.081
	*p*		<0.001	0.575	0.301
responsibility-related concerns	*r*	0.650	1	–0.165	0.053
	*p*	<0.001		0.027	0.495
infection-related concerns	*r*	0.042	–0.165	1	0.353
	*p*	0.575	0.027		<0.001
general pandemic-related concerns	*r*	0.081	0.053	0.353	1
	*p*	0.301	0.495	<0.001	

**Table 4 animals-12-01892-t004:** Mean differences of the groups “Long Covid” and “No infection”.

*Mean differences of the groups “Long Covid” and “No Infection” regarding symptoms of depression*									
BDI-II	**Long Covid (n = 65)**		**No Infection (n = 101)**					**95% CI**	
	M	SD	M	SD	*d*	t(166)	*p*	LL	UL
	35.07	9.18	28.33	6.71	0.731	4.4	<0.001	3.73	9.74
*Mean differences of the groups “Long Covid” and “No Infection” regarding the subjective experienced quality of life*									
WHOQOL-BREF Overall Score	**Long Covid (n = 83)**		**No Infection (n = 108)**					**95% CI**	
	M	SD	M	SD	*d*	t(191)	*p*	LL	UL
	55.44	15.19	75.28	11.7	−1.29	−8.5	<0.001	−24.50	−15.17
WHOQOL-BREF Physical Health	**Long Covid (n = 89)**		**No Infection (n = 111)**					**95% CI**	
	M	SD	M	SD	*d*	t(200)	*p*	LL	UL
	44.09	21.9	77.96	14.15	−1.60	−10.67	<0.001	−40.14	−27.6
WHOQOL-BREF Psychological Health	**Long Covid (n = 86)**		**No Infection (n = 112)**					**95% CI**	
	M	SD	M	SD	*d*	t(189)	*p*	LL	UL
	51.82	18.77	69.5	16.03	−0.896	−5.9	<0.001	−23.56	−11.79
WHOQOL-BREF Social Realtionships	**Long Covid (n = 88)**		**No Infection (n = 111)**					**95% CI**	
	M	SD	M	SD	*d*	t(199)	*p*	LL	UL
	59.22	23.83	72.82	19.83	−0.589	−3.63	<0.001	−20.99	−6.20
WHOQOL-BREF Environment	**Long Covid (n = 87)**		**No Infection (n = 109)**					**95% CI**	
	M	SD	M	SD	*d*	t(196)	*p*	LL	UL
	67.76	14.39	81.34	11.52	−0.750	−6.06	<0.001	−18.00	−9.15
*Mean differences of the groups “Long Covid” and “No Infection” regarding loneliness*									
Loneliness Scale	**Long Covid (n = 90)**		**No Infection (n = 97)**					**95% CI**	
	M	SD	M	SD	*d*	t(187)	*p*	LL	UL
	49.86	3.85	49.36	2.91	0.155	1.06	0.290	−0.43	1.44
*Mean differences of the groups “Long Covid” and “No Infection” regarding the subjectiv evaluated social support*									
ESSI	**Long Covid (n = 94)**		**No Infection (n = 96)**					**95% CI**	
	M	SD	M	SD	*d*	t(190)	*p*	LL	UL
	19.9	4.17	21.34	3.78	−0.398	−2.49	0.014	−2.57	−0.29
*Mean differences of the groups “Long Covid” and “No Infection” regarding the strength of the human-animal relationship*									
LAPS	**Long Covid (n = 79)**		**No Infection (n = 76)**					**95% CI**	
	M	SD	M	SD	*d*	t(155)	*p*	LL	UL
	75.49	9.07	72.52	9.57	0.318	1.98	0.049	0.08	5.92
*Mean differences of the groups “Long Covid” and “No Infection” regarding animal related concerns*									
animal-related concerns	**Long Covid (n = 87)**		**No Infection (n = 73)**					**95% CI**	
	M	SD	M	SD	*d*	t(160)	*p*	LL	UL
	1.58	0.83	2.05	1.02	−0.488	−3.16	0.002	−0.75	−0.17
responsibility-related concerns	**Long Covid (n = 86)**		**No Infection (n = 75)**					**95% CI**	
	M	SD	M	SD	*d*	t(161)	*p*	LL	UL
	1.67	0.92	2.24	0.99	−0.575	−3.74	<0.001	−0.87	−0.26
*Mean differences of the groups “Long Covid” and “No Infection” regarding pandemic related concerns*									
infection-related concerns	**Long Covid (n = 99)**		**No Infection (n = 96)**					**95% CI**	
	M	SD	M	SD	*d*	t(195)	*p*	LL	UL
	3.83	1.15	2.76	1.35	0.709	5.94	<0.001	0.71	1.42
general pandemic-related concerns	**Long Covid (n = 92)**		**No Infection (n = 93)**					**95% CI**	
	M	SD	M	SD	*d*	t(185)	*p*	LL	UL
	3.07	0.91	2.49	0.81	0.594	4.58	<0.001	0.33	0.83

Note. CI = Confidence Interval. LL = Lower Limit. UL = Upper Limit.

**Table 5 animals-12-01892-t005:** Significant correlations regarding pet-related concerns “Long Covid”.

*Significant correlations regarding animal-related concerns*		
**animal-related concerns**	** *r* **	** *p* **
WHOQOL-BREF Overall Score	−0.254	0.034
WHOQOL-BREF Environment	−0.253	0.031
WHOQOL-BREF Social Relationships	−0.232	0.045
*Significant correlations regarding responsibility-related concerns*		
**responsibility-related concerns**	** *r* **	** *p* **
WHOQOL-BREF Overall Score	−0.258	0.032

**Table 6 animals-12-01892-t006:** Significant correlations regarding pet-related concerns “No Infection”.

*Significant correlations regarding animal-related concerns*		
**animal-related concerns**	** *r* **	** *p* **
LAPS	0.337	0.004
*Significant correlations regarding responsibility-related concerns*		
**responsibility-related concerns**	** *r* **	** *p* **
LAPS	0.318	0.007

**Table 7 animals-12-01892-t007:** Significant correlations regarding pandemic related concerns “Long Covid”.

*Significant correlations regarding infectionl-related concerns*		
**infection-related concerns**	** *r* **	** *p* **
WHOQOL-BREF Overall Score	–0.282	0.018
WHOQOL-BREF Psychological health	–0.340	0.003
WHOQOL-BREF Physical Health	–0.409	<0.001
BDI-II	0.367	0.006
*Significant correlations regarding general pandemic-related concerns*		
**general pandemic-related concerns**	** *r* **	** *p* **
WHOQOL-BREF Overall Score	–0.370	0.003
WHOQOL-BREF Psychological health	–0.294	0.016
WHOQOL-BREF Environment	–0.459	<0.001
WHOQOL-BREF Social Relationships	–0.301	0.012
BDI-II	0.312	0.026
Loneliness Scale	0.331	0.004

**Table 8 animals-12-01892-t008:** Significant correlations regarding pandemic related concerns “No Infection”.

*Significant correlations regarding infectionl-related concerns*		
**infection-related concerns**	** *r* **	** *p* **
WHOQOL-BREF Physical Health	–0.262	0.035
Loneliness Scale	0.295	0.027
*Significant correlations regarding general pandemic-related concerns*		
**general pandemic-related concerns**	** *r* **	** *p* **
WHOQOL-BREF Overall Score	–0.504	<0.001
WHOQOL-BREF Psychological health	–0.521	<0.001
WHOQOL-BREF Environment	–0.422	<0.001
WHOQOL-BREF Social Relationships	–0.324	0.010
WHOQOL-BREF Physical Health	–0.326	0.009
ESSI	–0.377	0.005
BDI-II	0.512	<0.001
Loneliness Scale	0.338	0.012

**Table 9 animals-12-01892-t009:** Significant correlations regarding the strength of the human-animal relationship “Long Covid”.

LAPS	*r*	*p*
WHOQOL-BREF Overall Score	–0.321	0.010
WHOQOL-BREF Psychological health	–0.346	0.005
BDI-II	0.307	0.034

**Table 10 animals-12-01892-t010:** Deductive categorization of open question regarding concerns during the COVID-19 pandemic.

Category		Definition	Examples
Main Category	Subcategory		
Society	social division	concerns relating to the division of society	“division of society”; “opponents of vaccination are ostracized”
	economy and jobs	concerns regarding the economic situation and changes in the labor market	“that unemployment is rising more and more (gastronomy)”
	restrictions imposed by fundamental rights and freedoms	concerns regarding restrictions on own rights (fundamental rights and freedoms)	“fear of compulsory vaccination”; “restrictions on my freedom”
	other concerns related to social relations	concerns regarding the behavior of others	“the stupidity of people”; “lack of political action”
Health	own health—physiological	concerns regrading physical limitations	“Not being as fit afterwards or not being able to do sports so well”
	own health—psychological	concerns regarding psychological limitations	“Deterioration of my mental health”
	own contagion	concerns regarding own contagion	“Reinfection after work entry”; “With the infection I worry”
	health of related persons	concerns regarding health of familiy members and related personen	“That my daughter will never get better (Longcovid for one year).”
	contagion of related persons	concerns regarding contagion of family members and related persons	“Worry about family getting infected”
Daily Life	work, school, studies	concerns related to work, school or studies	“Fear of missing out on certain opportunities such as typical student life”
	financial situation	concerns regarding the financial situation in the houshold	“Financial shortages”
	living together in the family	concerns related to the life together in the family	“Quarrel in the family”
	housing situation	concerns regrading to the housing situation	“Changes in the living area”
	other everyday problems	concerns related to other everyday problems	“Lack of social contacts”; “not being able to do any more activities”

**Table 11 animals-12-01892-t011:** Deductive categorization regarding the pets role during the pandemic and when one is suffering from Long Covid condition.

Category	Definition	Characteristics	Examples
Biological impact	statements reffering to a biological impact of pet(s) on coping with Long-Covid	positive	“likewise, this helps with acute headaches, alone when I stroke the cat”; “walking the dog helps me to relax”
		neutral	“resaon to get some fresh air”; “relaxation”
		negative	“Much more effort to go for a walk”; “it is exhausting to go for a walk with the two twice a day even if you have no power”
Psychological impact	statements reffering to a psychological impact of pet(s) on coping with Long-Covid	positive	“Dog and cat calm me down, often they took away my stress and improved my mental condition”; “Intuitively my dog feels how I am and gives comfort”
		neutral	“comforting”
		negative	“Partial overload because the dogs have to go out, no matter how lousy you feel mentally”; “I am more quickly annoyed by them”; “worries if i can take care of my pet in the future”
Social impact	statements reffering to a social impact of pet(s) on coping with Long-Covid	positive	"The relationship with my pet has improved”; “I find it easier to be alone at home because another living being is there”
		neutral	“contact”

## Data Availability

Not applicable.

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
