# Peer review of "Fantastic Beasts and Why It Is Necessary to Understand Our Relationship—Animal Companionship under Challenging Circumstances Using the Example of Long-Covid"

_animals, 2022, doi:10.3390/ani12151892_

Round 1
Reviewer 1 Report
I would suggest another read through with minor grammatical corrections, looking out for places in which commas can be added in long sentences to enhance readability. Other small edits I noted: line 62- "believe" should be "belief". Lines 198-200 are unclear to me. Table 10 runs off the document from my view.
Reviewer 2 Report
Thank you for submitting your paper about an interesting and timely topic. I find it well written. The methods and discussion in particular are quite clear.
I believe that there are a few places in the paper where there is an incorrect word - Line 87 heir should be there, Line 121 resistance should be residence - so please have a close look at that. And, there are a few places where the word choice could be changed to be a bit clearer. Perhaps use the function in word processing programs that reads the paper aloud, I find that helpful with my papers. Or ask a colleague who does not know your work well to have a close read.
I suggest adding a bit of background about the pet effect. Maybe reference Vormbrock 1988 J Beav Med 11(5) 509-517 or Allen 2003 Curr Dir in Psychol Sci 12 236-239? Adding that would give better context for the pet effect paradox. When referencing the pet effect paradox, perhaps consider adding Herzog's 2020 and 2022 Psychology Today columns about them to the reference list also.
Line 168 - should Enrichd be all capital letters? I believe it is an acronym.
Line 188 and 200 - is there an example statement you could add to make this more consistent with the instruments above?
Line 208 - format example as others were formatted for consistency and ease to reader.
Regarding your references, with the exception of the instruments or methods, they are all quite new. Are these the best foundational references available? It can be tempting to use the most recent and easily at hand references when perhaps more foundational would help the reader.
